# The Flagellar Transcriptional Regulator FtcR Controls *Brucella melitensis* 16M Biofilm Formation via a *betI*-Mediated Pathway in Response to Hyperosmotic Stress

**DOI:** 10.3390/ijms23179905

**Published:** 2022-08-31

**Authors:** Jia Guo, Xingmei Deng, Yu Zhang, Shengnan Song, Tianyi Zhao, Dexin Zhu, Shuzhu Cao, Peter Ivanovic Baryshnikov, Gang Cao, Hugh T. Blair, Chuangfu Chen, Xinli Gu, Liangbo Liu, Hui Zhang

**Affiliations:** 1State International Joint Research Center for Animal Health Breeding, College of Animal Science and Technology, Shihezi University, Shihezi 832003, China; 2College of Veterinary, Altai State Agricultural University, 656000 Barnaul, Russia; 3State Key Laboratory of Agricultural Microbiology, Huazhong Agricultural University, Wuhan 430000, China; 4International Sheep Research Center, School of Agriculture and Environment, Massey University, Palmerston North 4442, New Zealand

**Keywords:** *Brucella melitensis* 16M, flagellar transcriptional factor FtcR, *betI*, biofilm, hyperosmotic stress

## Abstract

The expression of flagellar proteins in *Brucella* species likely evolved through genetic transference from other microorganisms, and contributed to virulence, adaptability, and biofilm formation. Despite significant progress in defining the molecular mechanisms behind flagellar gene expression, the genetic program controlling biofilm formation remains unclear. The flagellar transcriptional factor (FtcR) is a master regulator of the flagellar system’s expression, and is critical for *B. melitensis* 16M’s flagellar biogenesis and virulence. Here, we demonstrate that FtcR mediates biofilm formation under hyperosmotic stress. Chromatin immunoprecipitation with next-generation sequencing for FtcR and RNA sequencing of *ftcR*-mutant and wild-type strains revealed a core set of FtcR target genes. We identified a novel FtcR-binding site in the promoter region of the osmotic-stress-response regulator gene *betI*, which is important for the survival of *B. melitensis* 16M under hyperosmotic stress. Strikingly, this site autoregulates its expression to benefit biofilm bacteria’s survival under hyperosmotic stress. Moreover, biofilm reduction in *ftcR* mutants is independent of the flagellar target gene *fliF*. Collectively, our study provides new insights into the extent and functionality of flagellar-related transcriptional networks in biofilm formation, and presents phenotypic and evolutionary adaptations that alter the regulation of *B. melitensis* 16M to confer increased tolerance to hyperosmotic stress.

## 1. Introduction

The *Brucella* genus contains Gram-negative and intracellular bacteria [1,2] that are listed as category B bioterrorism agents [3], rendering them a considerable concern to human and environmental welfare. Animal and human brucellosis show similar pathological phenomena at the cellular level [4]. These pathogens use lipopolysaccharides, β-1,2-D-glucans, T4SS effectors, and other virulence factors to prevent phagosome–lysosome fusion and escape to the endoplasmic reticulum to develop a replicative niche within phagocytic and other non-antigen-presenting cells, producing a systemic infection [4,5,6,7]. Brucellosis infection is characterized by undulant fever, night sweats, and arthralgia in humans [8,9]. In cattle, this infection primarily results in abortion and infertility [10].

*Brucella* infections are among the most difficult to treat. In addition to their capacity to survive in phagocytic cells [5], *Brucella* species develop a community of microbial cells that aggregate within an extracellular matrix [11,12], known as a biofilm. Biofilms are formed by a complex process that initiates when microorganisms attach to a surface via an extracellular matrix and other adhesins that allow them to develop into large communities [13]. Compared to planktonic bacteria, biofilms have unique transcriptional and expressive programs [14,15]. Bacteria form a niche within these biofilms, exhibiting increased virulence and resistance to various hostile microenvironments [16,17]. It has been reported that *Brucella* can form a biofilm or aggregate in response to desiccation [11]. Moreover, *Brucella* biofilms are associated with chronic infection during their life cycle [18]. As a result, biofilms are difficult to treat, and are important potential agents of chronic infectious diseases [19]. It has been well established that the flagellum plays an essential role in the adaptation and survival of cells during the formation of biofilms [13,20,21]. Therefore, strategies for controlling and treating biofilm-related infections can be improved through a better understanding of the role of flagella in bacterial biofilms’ formation and development.

The underlying molecular mechanisms through which the flagella of *B. melitensis* 16M regulate biofilm formation are poorly understood. It is known that the quorum-sensing system (VjbR) is involved in expressing the flagellar transcriptional regulator (FtcR) in *B. melitensis* 16M [12,22], and FtcR was shown to regulate flagellar activation and expression throughout the life cycle of *B. melitensis* 16M [22]. Interestingly, the *B. melitensis* 16M Δ*ftcR* strain was defective in the production of FlgE and FliC flagellar proteins, and *fliF* promoter activity was reduced during vegetative growth compared to that of the wild-type (WT) strain [22]. Moreover, observing the Δ*ftcR* strain using electron microscopy revealed a major defect in the flagellar structure under growth conditions in which the WT strain was flagellated, suggesting that FtcR is a key regulator required for switching flagellar gene expression in *Brucella* [22,23,24]. Genomic sequencing of isolated, non-motile *Brucella* has detected nonfunctional flagellar genes [24]. In contrast, an analysis of genes encoding flagellar proteins in a motile strain revealed that all genes were fully functional [25], emphasizing that some genes may be expressed under environmental selection pressure and perform specific biological functions [26,27]. Previous studies showed that flagellar genes could be activated in response to environmental changes or stress [28,29]. Moreover, comparative sequence analysis in *B. melitensis* 16M showed that ActR—a protein involved in regulating flagellar and biofilm formation—shared 53.2% genetic similarity with FtcR [30]. We therefore suspect that FtcR-like regulators could be involved in flagellar biosynthesis and biofilm formation.

Here, we used an integrative approach, combining chromatin immunoprecipitation assays with next-generation sequencing (ChIP–Seq), transcriptomics, and microscopy to characterize the biological functions of FtcR and investigate the changes in the genetic molecular mechanisms during *B. melitensis* 16M biofilm formation. Our data suggest for the first time that FtcR is a new global transcriptional regulator in *B. melitensis* 16M that regulates the expression of flagellar genes and the hyperosmotic-stress-response regulator during the formation of biofilm cells under these conditions.

## 2. Results

### 2.1. FtcR Is Required for B. melitensis 16M’s Survival in Hostile Environments in Planktonic and Biofilm States

The intricacies of flagellar gene expression in *B. melitensis* 16M are tied to its life cycle. We established an intracellular model by simulating a ∆*ftcR* infection in RAW264.7 macrophages. After 24 h from initiating the infection, the Δ*ftcR* strain did not differ from the WT strain (Figure 1A). In general, *ftcR* was not essential for the short-term intracellular survival of *B. melitensis* 16M in macrophages. Nevertheless, it has been suggested that *ftcR* plays a key role in chronic and stealthy *Brucella* infections [22,31]. As a result of pathogen–host interactions, bacteria are exposed to various physiological and biological stresses [32,33]; thus, it is tempting to hypothesize that FtcR may have unknown functions in response to hostile environments.

When analyzing the response of Δ*ftcR* to various environmental cues under consistent nutritional conditions, our results show that the survival rate of ∆*ftcR* decreased under heat shock, osmotic stress, H_2_O_2_, and polymyxin stress conditions compared to that of the WT strain (Figure 1B), but was partially or fully restored in the ∆*ftcR* complemented strain, ∆*ftcR*-C. Therefore, FtcR is required for the response to hostile environmental stress conditions in *B. melitensis* 16M in planktonic states. Next, we assessed the contribution of *ftcR* to biofilm formation under stress. To examine whether the biofilms formed by Δ*ftcR* could resist a range of hyperosmotic environmental stresses, we improved the *B. abortus* biofilm development method established by Tang et al. [14]. First, we grew biofilms under various stress conditions for 20 d. After normalizing the biomass to the untreated control of each strain, we identified three strains with similar relative reductions in biomass after treatment with the osmotic solute. We found that Δ*ftcR* biofilms were significantly more susceptible to osmotic solutes than those of the WT strain (Figure 1C,D). To further test whether the observed growth defects were due to hyperosmotic stress, we investigated different types of osmotic solutes to rule out responses due to high concentrations of NaCl. Under different hyperosmotic stress treatments (i.e., NaCl, KCl, sucrose, and dextran), the Δ*ftcR* strain exhibited the same phenotypic changes as those observed in the NaCl when compared with those of the WT strain (Appendix A). Moreover, the *ftcR* complemented strain restored the mutant phenotype. These results demonstrate that the inactivation of *ftcR* significantly affects the physiology and survival of *B. melitensis* 16M following hyperosmotic challenges.

### 2.2. Scriptional Autoregulatory Properties of FtcR and Indepess

To more thoroughly define the biofilms’ sensitivity to hyperosmotic stress and more closely approximate their growth stage, we grew the biofilms under 1.5 M NaCl for 20 d, and quantified their biomass every 2 d (Appendix A). The Δ*ftcR*, WT, and Δ*ftcR*-C treated strains did not show significant changes during the first 4 d (Appendix A). After day 4, Δ*ftcR* showed a greater biomass reduction relative to the other two strains, and this difference was most pronounced after 10 d (Appendix A). After 20 d of static culture, confocal laser scanning microscopy (CLSM) showed significant disruptions in the biofilms of the Δ*ftcR*, WT, and Δ*ftcR*-C strains treated with hyperosmotic stress relative to those left untreated (Figure 2A–C). However, the biofilms of the Δ*ftcR* strain exhibited weakened tolerance to hyperosmotic stress (Figure 2A–C), as evidenced by the biomass, which was 3.12 μm^3^/μm^2^ on average (Figure 2A,B), with an average and maximum biofilm thickness of 2.17 and 4.31 μm, respectively (Figure 2A,C), and appeared to be composed of threefold less biomass than the biofilms formed by WT under the same hyperosmotic stress. We obtained consistent results with crystal violet (CV) staining (Appendix A). These data indicate that the hyperosmotic stress treatment induced a significant loss of biomass in Δ*ftcR* biofilms compared with the biofilms formed by the WT and complemented mutant strains. To determine whether the hyperosmotic-stress-induced biomass reductions were caused by increased cell death, we quantified the survival of biofilm cells after 20 d of hyperosmotic stress treatment in a colony biofilm model. Compared to the WT, the Δ*ftcR* biofilms showed significantly reduced cell survival after treatment with 1.5 M NaCl (Figure 2D), indicating that the Δ*ftcR* biofilm cells were more sensitive to 1.5 M NaCl-induced cell death than the WT biofilms. These results suggest that *ftcR* mediates the hyperosmotic response in *B. melitensis* 16M biofilms.

### 2.3. Identification of FtcR Target Genes Using RNA-Seq and ChIP-Seq Analyses

FtcR is predicted to function as a DNA-binding response regulator that contains receiver (REC) and winged-helix (wHTH) domains [22]; however, the direct targets of FtcR have yet to be precisely determined. To gain insight into the hyperosmotic stress signaling regulatory pathway and the key target genes of FtcR involved in biofilm growth, a transcriptome sequencing (RNA-Seq) experiment was conducted to profile the transcripts of the *B. melitensis* 16M biofilms formed by the Δ*ftcR* and WT strains. Based on their expression levels, cluster analysis was performed to arrange the samples into groups to elucidate possible relationships (Figure 3A). Our comparative transcriptomic analysis revealed 564 differentially expressed genes (DEGs) (adjusted *p* (Padj) < 0.05, fold change ≥ 1), of which 223 genes were upregulated and 341 were downregulated. A volcano plot is shown in Figure 3B. Based on the Kyoto Encyclopedia of Genes and Genomes (KEGG) pathway enrichment analysis, most of these genes are involved in flagellar assembly, ribosome, and bacterial secretion systems (Figure 3C). We performed a Gene Ontology (GO) enrichment analysis to explore the different biological processes of the biofilms between the Δ*ftcR* and WT strains. Establishment of localization, transport, and localization were the dominant groups in all three DEG sets (Figure 3D).

RNA-Seq revealed the gene expression profile of FtcR, and ChIP was then performed to identify binding sites for FtcR. On chromosomes I and II, there were 363 binding sites for FtcR (Figure 4A), which were enriched near transcription start sites (Figure 4B). RNA-Seq combined with ChIP-Seq identified 54 regulon target genes of FtcR, involved in transporting and binding proteins, regulatory functions, cell envelopes, translation, hypothetical proteins, fatty acid and phospholipid metabolism, cellular processes, and energy metabolism, were identified in the ChIP-Seq data (Figure 4C and Appendix A). Additionally a heatmap of the 54 regulon target genes was generated to visualize the DEGs (Figure 4D). It will be necessary to explore in further detail the functions of the genes that FtcR directly regulates to shed light on the mechanisms of the *B. melitensis* 16M biofilm under hyperosmotic stress.

### 2.4. Transcriptional Autoregulatory Properties of FtcR and Independent of FliF

Using ChIP-Seq, we found significant enrichment of *ftcR* in the cognate promoter region (Figure 5A), and to identify FtcR-bound DNA motifs, multiple expectation maximization was used for motif elicitation of the identified expression peaks. An 8 bp TGTGGGCA-binding region was identified as having the highest E value (1 × 10^−9^) and frequency (118/363) of FtcR DNA-binding motifs (Figure 5B). Furthermore, the electrophoretic mobility shift assays (EMSAs) confirmed that FtcR binds to the motif-containing promoter fragments of *ftcR* in vitro (Figure 5C). The specificity of this protein–DNA interaction was highlighted when excess unlabeled DNA or mutation of the motif in the *ftcR* fragment blocked its interaction with the promoter fragments (Figure 5D). Using the *ftcR* promoter–*lacZ* transcriptional fusion plasmid, reporter assays were conducted to determine whether FtcR affects gene expression from the *ftcR* promoter. We constructed the *ftcR* promoter–*lacZ* plasmid and transformed it into the WT and ∆*ftcR* strains; the results showed that FtcR autoregulated the transcriptional activity of its cognate promoter under hyperosmotic conditions, unlike the WT strain (Figure 5E); moreover, although FtcR appeared to autoregulate its expression under hyperosmotic stress (Appendix A), excess solute amounts inhibited FtcR expression (Appendix A). In summary, our findings indicate that FtcR specifically binds to the region upstream of *ftcR* in response to hyperosmotic stress, thereby promoting *ftcR* expression.

Leónard et al. [22] defined FtcR as a flagellar master regulator in *B. melitensis* 16M that binds directly to the upstream region of the *fliF* gene, which is consistent with our transcriptome and ChIP-Seq analysis showing that FtcR was specifically involved in regulating the transcription of *fliF* (Appendix A). Since *fliF* promotes biofilm formation, we investigated whether FtcR-mediated flagella are involved in the hyperosmotic regulation of biofilms. Thus, we focused on the role of FliF in biofilms. 

To determine whether the FliF protein was produced in the biofilm state as part of the biomass of biofilms in response to changes under hyperosmotic stress, we performed Western blot (WB) analysis on whole-cell extracts from *B. melitensis* 16M at the exponential phase of growth and the biofilm stage. Specific antisera to FliF allowed us to visualize the expression of FliF monomers at the exponential phase of growth (Appendix A). This protein was not detected at the biofilm stage (Appendix A). In addition, the biomass of the Δ*fliF* strain did not differ from that of the WT under hyperosmotic stress treatment (Appendix A). We used LIVE/DEAD BacLight bacterial fluorescence cell viability kits to monitor bacterial viability in *B. melitensis* 16M biofilms to distinguish bacteria with intact cell membranes (which appear green in color) from those with damaged membranes (in red). Compared with the WT, the Δ*fliF* strain did not develop obvious osmotic-induced defects (Appendix A). Fluorescence-activated cell sorting (FACS) analysis of 50 μm filtrates was used to accurately measure bacterial viability in the aggregated cell populations. The results did not indicate obvious defects in the Δ*fliF* strain compared with the WT (Appendix A). Thus, *B. melitensis* 16M biofilm formation under hyperosmotic stress was independent of *fliF*.

### 2.5. FtcR Mediates the Transport Pathway in B. melitensis 16M Biofilms under Hyperosmotic Stress

Previous studies have indicated that transporter proteins play a central role in metabolic and energy-producing pathways in bacteria that respond to osmotic stress [34,35]. We found that the most of the genes differentially regulated by FtcR in biofilm cells encoded transport and binding proteins, suggesting that FtcR mediates the regulation of transport during the development of *B. melitensis* 16M biofilms. The gene expression changes related to transporter pathways were assessed to explore whether these pathways respond to hyperosmotic stress mediated by FtcR in the *B. melitensis* 16M biofilm. Specifically, upregulated genes involved in transport included *BME_RS07095*, *BME_RS06870*, *BME_RS09570*, *BME_RS02235*, *BME_RS14635*, *BME_RS05400*, *sbmA*, and *BME_RS01300* (Figure 6A), while downregulated genes included *amtB*, *betI*, *BME_RS00805*, *BME_RS06800*, and *BME_RS12810* (Figure 6A). The expression of these genes, as determined by qRT-PCR, coincided with the RNA-Seq results (Figure 6A,B). Notably, *betI* gene expression was significantly reduced in Δ*ftcR* (Figure 6A,B). The BetI protein has been reported to participate in osmotic regulation as a choline sensor, acting as a transcriptional repressor of the *betIBA* operon, *betT1*, and *betT2*, which are induced by salt treatment to adapt to osmotic stress [36,37,38,39]. Thus, we speculated that *betI*, which FtcR directly regulates, is involved in the regulation of hyperosmotic stress in *B. melitensis* 16M biofilms.

### 2.6. FtcR Binds to the Promoter of betI in Response to Hyperosmotic Stress Environments

We investigated whether *betI* is directly regulated by FtcR in hyperosmotic responses. ChIP-Seq results showed a significant enrichment of *betI* at the promoter region (Figure 7A). The in vitro binding of FtcR to motif-containing fragments of *betI* was also confirmed by EMSAs (Figure 7B). An excess of unlabeled DNA or mutation of the motif in the *betI* fragment prevented the interaction between FtcR and the promoter fragment, illustrating the specificity of the interaction between FtcR and DNA (Figure 7C). Furthermore, the RNA-Seq data showed that *betI* expression decreased, which was also consistent with the WB data (Figure 7D).

To determine whether FtcR regulates the biofilms’ hyperosmotic stress response through *betI*, we generated *betI* mutant strains to explore their biological functions in response to hyperosmotic stress. The results show that compared to WT, Δ*betI* had a reduced tolerance to hyperosmotic stress during the biofilm stage; the average biomass was 2.11 μm^3^/μm^2^ (Figure 8A,B), while the average and maximum biofilm thickness were 1.14 and 4.68 μm (Figure 8A,C), respectively. Moreover, the biofilms formed by Δ*betI* appeared to be composed of threefold less biomass than the biofilms formed by the WT under the 1.5 M NaCl treatment. We obtained consistent results with CV staining and bacterial plate counts (Appendix A and Figure 8D). According to the results, *betI* plays an important role in maintaining bacterial survival in biofilms under hyperosmotic conditions.

## 3. Discussion

The flagella of *Brucella* species contribute considerably to establishing the pathogens’ replicative niche, chronicity of infection, and ability to form biofilms [21,40]. However, the underlying molecular mechanisms of *Brucella*’s flagella and biofilm formation are poorly understood. The objective of this research was to explore the potential mechanisms through which the flagella affect biofilm formation, focusing on the FtcR protein of *B. melitensis* 16M. For the first time, we showed that FtcR is a regulator of biofilms’ responses to environmental hyperosmotic stress. A combination of RNA-Seq and ChIP-Seq analyses revealed that FtcR directs the expression of multiple potential target genes, including *ftcR* and *betI*. FtcR is an OmpR homolog that autoregulates its expression at the transcriptional level during hyperosmotic stress at the biofilm stage, and this function is also likely achieved through the positive regulation of the osmotic-stress-response regulator BetI, which maintains cell survival and function under hyperosmotic stress. Our findings demonstrate that FtcR is a global regulator that plays an essential role in biofilm responses to hyperosmotic stress, promoting considerable changes in the transcriptome.

Regulation of intracellular survival is a functional phenotype associated with the flagella in *Brucella*. This mechanism includes a lysosomal degradation pathway that allows intracellular survival and replication in eukaryotic cells [21]. Our results show that the inactivation of *ftcR* did not alter *Brucella*’s survival in host macrophage cells after 24 h. The *ftcR* genes were not involved in *B. melitensis* 16M intracellular trafficking, potentially because of the high energy cost required for their synthesis and assembly to support *Brucella*’s replicative niche compared with periods of chronic infection [31,40]. However, the *ftcR* mutant was attenuated in BALB/c mice, which is consistent with data obtained from *Brucella* strains with mutations in genes encoding different structural flagellar components [22]. Nevertheless, other mechanisms are needed to explain the attenuation observed after four weeks and at later time points. Persistent intracellular bacterial infection results from the combined effects of bacterial adaptations under selection pressure [41,42]. Given that the genes in OmpR family are involved in stress adaptation and virulence [43], it is tempting to hypothesize that FtcR may have unknown functions in response to hostile environments—especially the long-term hostile environments of the *Brucella* vacuole [4]. We investigated the responses of *ftcR* mutants to various environmental cues, and found that the FtcR mutant has a reduced survival rate under heat shock, osmotic stress, H_2_O_2_, and polymyxin stress conditions. Thus, the function of the OmpR family regulator response is to maintain survival under stress. These results are consistent with those of previous studies [43].

Under unfavorable conditions for the bacteria, the biofilm plays an essential role in bacterial survival and growth, and is regulated in response to changing environmental conditions [44,45]. We investigated the role of FtcR in the response of *B. melitensis* 16M biofilms to various conditions of environmental stress. Inconsistent with *B. melitensis* 16M in a planktonic state, the *ftcR* mutant showed a significant biofilm growth defect compared with that of the WT and complemented strains only under hyperosmotic stress; *B. melitensis* 16M did not show the same susceptibility to other stressors as the planktonic bacteria of the Δ*ftcR* strain, possibly highlighting the difference in the protective effects of the biofilm structure on bacteria and the bacteria in the planktonic state [11,14,15]. This biofilm phenotype was not a response to high concentrations of NaCl alone, because it was also observed when the biofilms were treated with other types of osmotic solutes. To gain further insights into the mechanisms behind the FtcR-mediated biofilm formation under hyperosmotic stress, we performed comparative transcriptomics between biofilms formed by Δ*ftcR* and WT, and combined it with ChIP-Seq to reveal 54 bona fide core regulons controlled by FtcR. Thus, FtcR is probably an OmpR family transcriptional regulator that affects a wide range of genes, based on these results.

Several organisms, including *Escherichia coli*, *Shigella*, *Salmonella*, and *Yersinia*, have been found to express OmpR [46,47,48,49]. This regulator controls target genes in response to changes in osmolarity, pH, and temperature by controlling cellular adhesion, motility, and temperature [50,51,52]. The RNA-Seq and ChIP-Seq analyses found that FtcR autoregulates its expression. The DEGs in the RNA-Seq data were associated with flagellar assembly, which is consistent with the findings of the present study. Moreover, FtcR is a master regulator of the flagellar system of *B. melitensis* 16M [22], and binds directly to the upstream region of *fliF*. Historically, mature biofilms were thought to turn off gene expression encoding flagella [53,54]. However, other studies propose that the flagella play a structural role, acting to hold cells together as well as binding them to surfaces [55,56]. We examined the role of the major downstream target gene *fliF* in biofilms, and showed that our results were consistent with the current state of the research in that this flagellar gene was expressed only in the logarithmic growth phase, and subsequently disappeared. Furthermore, *fliF* was not involved in the hyperosmotic stress response of *B. melitensis* 16M biofilms. Attenuation of the *filF* mutant in our study leads us to speculate that the flagellar genes are expressed under several specific conditions encountered during the infectious cycle [40,57]. Their role in biofilms remains to be further verified.

According to a systematic transcriptome analysis of all two-component regulatory systems, the OmpR system was also shown to be involved in amino acid metabolism and transport in *E. coli* [58]. Additionally, analysis of transport and binding DEGs between biofilms formed by Δ*ftcR* and WT also showed differences in our study, and highlighted the importance of FtcR in regulating transport and binding proteins in the biofilms under hyperosmotic stress. It is possible to counterbalance osmotic differences through the accumulation of compatible solutes, such as polyols, sugars, amino acids and their derivatives, and betaines. Both *E. coli* and *B. subtilis* are known to accumulate and synthesize more compatible solutes than they can synthesize independently [59]. Thus, physiological protection from stress is likely attributed to the stabilizing effect of compatible solutes on macromolecules and biosynthesis processes. The ABC transporter family functions as a high-affinity uptake system for compatible solutes, having the most substantial protective effect on cell growth when exposed to high salinity [60]. We speculate that the FtcR-mediated *B. melitensis* 16M transport system may be involved in regulating hyperosmotic stress responses in biofilms. Our findings show that FtcR alters the patterns of regulation of the DEGs involved in binding and transporting proteins and, furthermore, that FtcR plays a direct role in regulating the gene expression of numerous transporters in response to changes in hyperosmotic stress, including the *betI* pathway, which has been demonstrated to be crucial for osmotic stress responses in bacteria [36]. In addition to acting as a choline sensor, BetI represses the transcription of BetIBA, BetT1, and BetT2. Notably, our results corroborate those obtained in *V. harveyi* [37], which found that BetI regulates hyperosmotic stress. BetI likely supports osmoprotection through the synthesis of the osmoprotectant glycine betaine. Additionally, BetI activates the expression of genes encoding regulatory small RNAs that control quorum-sensing transitions [37]. Thus, our findings reveal that FtcR controls BetI production to maintain the physiological status of *B. melitensis* 16M biofilms.

In summary, our results suggest that the decreased hyperosmotic stress tolerance of Δ*ftcR* biofilms is due to the inhibition of the osmotic-stress-response regulator BetI. In addition, altered global expression of genes involved in various metabolic processes of the biofilm can also contribute to hyperosmotic tolerance in Δ*ftcR* biofilms. Although the metabolic activity of the Δ*ftcR* biofilm requires further investigation, it is likely that the FtcR-mediated regulation of BetI helps maintain cell survival by supporting the biofilm during hyperosmotic stress. Furthermore, this pathway is independent of the downstream flagellar *fliF* gene controlled by FtcR (Figure 9). Our study demonstrates for the first time that the involvement of the flagellar transcriptional regulator protein FtcR in *B. melitensis* 16M biofilms increases tolerance to hyperosmotic stress. Consequently, this regulatory pathway should be evaluated in other *Brucella* species that have been discovered—especially those that are pathogenic to marine and terrestrial mammals. The involvement of FtcR in *B. melitensis* 16M biofilms, along with its link to genetic and biochemical programs required for adaptation and survival, may represent an active mechanism that supports the emergence of microcolony aggregates with increased hyperosmotic tolerance in biofilm cells, which assist in broadening their host range. While the involvement of *ftcR* in hyperosmotic tolerance remains to be investigated in other bacterial pathogens, the knowledge obtained in this work can help us to better understand the biofilm-specific tolerance and facilitate the development of counteracting strategies.

## 4. Materials and Methods

### 4.1. Bacterial Strains and Cell Lines

The primers of mutants and complementation used in this research are listed in Appendix A. All *Brucella* strains were grown at 37 °C using either BBL^TM^ Brucella Agar or BBL^TM^ Brucella Broth (Difco, Franklin Lakes, New Jersey, USA). *E. coli* was grown on Luria broth or on Luria broth agar plates at 37 °C. The pBBR1MCS4 vector was maintained in our laboratory. The *B. melitensis* 16M green fluorescent protein (GFP) was provided by the Xinjiang Center for Disease Control and Prevention. When necessary, kanamycin (20 μg/mL), ampicillin (20 μg/mL), and chloramphenicol (15 μg/mL) were added to the culture media. The strains containing the *ftcR*, *fliF*, and *betI* gene deletions (∆*ftcR*, ∆*fliF*, and ∆*betI*, respectively) were acquired via resistance gene replacement, as described previously [61]. The *ftcR*, *fliF*, and *betI* complemented strains (∆*ftcR-C*, ∆*fliF-C*, and ∆*betI-C*, respectively) were acquired as described previously [22]. The complementation vector pBBR1MCS4—a plasmid that can replicate in *Brucella*—was introduced into electrocompetent cells of ∆*ftcR*, ∆*fliF*, and ∆*betI* via electroporation. All work with *Brucella* strains was conducted in a biosafety level 3 laboratory. The murine macrophage cell line RAW264.7 (ATCC, TIB-71) was cultured in Dulbecco’s modified Eagle medium (Thermo Fisher Scientific, Shanghai, China) (from passage number 20 to passage number 30) supplemented with 10% fetal bovine serum (Taixin, China) at 37 °C with 5% CO_2_ (vol/vol).

### 4.2. Infection Assay

RAW264.7 cells were seeded 2 d prior to infection at 6 × 10^4^ cells/well in 24-well plates, and infected with *Brucella* strains at a multiplicity of infection (MOI) of 10. Infections were performed as described previously [62]. To quantify bacterial replication, coverslips were processed for immunofluorescence microscopy, and the number of intracellular bacteria was scored blinded in ~100 infected cells/experiment within random fields. Each experiment was repeated at least three times independently.

### 4.3. Stress Tolerance in Planktonic Cells

The *B. melitensis* 16M strains were evaluated for their sensitivity to environmental stress using an in vitro model, as previously described in [61,63]. The WT, ∆*ftcR*, and ∆*ftcR*-C strains were cultured until they reached the logarithmic growth phase, centrifuged at 13,000× *g* for 15 min, and then rinsed twice with phosphate-buffered saline (PBS). The bacteria were counted, and 10^8^ colony-forming units (CFUs) were cultured in 1 mL of Brucella Broth at 37 °C for 30 min under different stress conditions: 1.5 M NaCl, at pH 2.5, pH 11.5, and pH 7.0 (at 50 °C for 30 min); polymyxin B; and 10 mM H_2_O_2_. After centrifugation, the culture medium was discarded, and the bacteria were resuspended in 1 mL of PBS before being applied to solid medium plates in a 10-fold serial dilution. We then counted the CFUs and calculated the percentage of bacteria that survived this stress test (% survival), and compared it to the % survival measured in the WT and control strains (% survival).

### 4.4. Biofilm Culture and Observation

Biofilms were cultured in 24-well plates as described by Tang et al. [14] and Almirón et al. [11], with some modifications. Briefly, Brucella Broth was inoculated with a single colony and cultured at 37 °C and 5% CO_2_ with shaking at 150,000× *g* until growth reached the logarithmic phase, before being transferred (2 mL per well) into a 24-well cell culture plate with sterilized cover slips (8 × 8 mm, adhesion carrier). Next, the cells were incubated with 1.5 M NaCl for 20 d to induce high hyperosmotic stress; the culture medium was changed every 10 d. The attached carrier was dipped slowly into a small tub of water, shaken out, and the process was repeated to remove unattached bacterial components and reduce background staining. The biofilm formed by *Brucella* GFP was observed via CLSM; confocal images of the biofilm were analyzed using NIS-Elements Viewer 4.20 software (Nikon, Inc., Tokyo, Japan). The software reconstructed the two-dimensional intensity of fluorescence of all scanned layers into a three-dimensional volume stack at each cycle of scanning, and quantitative analysis of the biofilm images was performed using the Comstat2 program [64].

### 4.5. Biomass Assay

Overnight cultures of *Brucella* were seeded in Brucella Broth in borosilicate tubes for static incubation at 37 °C for 24 d; fresh medium was added every 5 d. The planktonic phase in the borosilicate tubes was removed. Then, 250 μL of 0.1% filtered CV staining solution was added to each well and incubated for 15 min. The staining solution was removed by inversion, and then the plate was washed three times with PBS before being fully dried in an incubator. Next, 200 μL of 33% glacial acetic acid solution was added to each well, and the borosilicate tubes were placed on a shaker to uniformly dissolve the dye solution before the OD_550_ was measured [65].

### 4.6. Biofilm Immunoblots

We washed the enriched biofilm samples twice with PBS, and then lysed them with DNase I before adding protease inhibitors (Takara, Beijing, China). The samples were then centrifuged at 15,000× *g* for 30 min at 4 °C, the supernatant was removed, and the protein concentration was determined using a BCA protein quantification kit (Thermo Fisher Scientific, Shanghai, China). The samples were mixed with SDS–PAGE loading buffer and boiled at 100 °C for 10 min in a water bath prior to SDS–PAGE in a 12% polyacrylamide gel. The proteins in the gel were transferred to a polyvinylidene fluoride membrane, which was blocked overnight in Tris-buffered saline containing 99% Tween-20 (TBS-T) and 5% milk. The membrane was washed three times with TBS-T, and FtcR-, FliF-, and BetI-specific antisera were added and incubated for 3 h. A monoclonal anti-Omp10 antibody was used as a loading control. After the membrane was washed three times for 5 min each time in TBS-T, an anti-rabbit IgG antibody conjugated with peroxidase was added at a dilution of 1:1000 for 3 h with shaking at 37 °C. The resulting bands were detected using ProteinSimple (FluorChemE, San Leandro, CA, USA). A graph of the relative protein expression levels was prepared using Image Lab 3.0 software (Bio-Rad, San Diego, CA, USA).

### 4.7. Chromatin Immunoprecipitation and High-Throughput Sequencing (ChIP-Seq)

The ChIP assay was performed as previously described, with some modifications [66]. The *B. melitensis* 16M strain (OD_600_ = 0.01) was grown in Brucella Broth medium to OD_600_ = 0.3 to induce expression of FtcR. Then, we resuspended the bacteria in precooled buffer (PBS, protease inhibitor, phenylmethylsulfonyl fluoride) and incubated them with formaldehyde for 10 min at 37 °C in a static mix. Next, 125 mM glycine was added to stop the crosslinking, and the samples were washed three times in ice-cold PBS. The bacterial cell pellets were resuspended in 500 μL of immunoprecipitation (IP) buffer (50 mM HEPES-KOH, 1 mM EDTA, 150 mM NaCl, 1% Triton X-100, 0.1% sodium deoxycholate, 0.1% SDS, and Complete Protease Inhibitor Cocktail), and chromatin was fragmented using a Bioruptor in an ice bath. A NEXTflex ChIP-Seq kit (Bioo Scientific, Austin, TX, USA) was used to construct the genome libraries. Protein G magnetic beads were washed in lysis buffer and then blocked with 8 mg/mL bovine serum albumin (BSA), before being incubated with the fragmented chromatin and a polyclonal rabbit anti-FtcR antiserum for 5 h at 4 °C on a rotator. The supernatants were treated with 10 mg/mL RNAse A (Sigma-Aldrich, Taufkirchen, Germany) for 1 h at 37 °C, followed by 0.2 mg/mL Proteinase K (Sigma-Aldrich, Taufkirchen, Germany) for 30 min at 55 °C, to stop the reaction. The DNA fragments were purified using a purification kit (Invitrogen, Carlsbad, CA, USA). The immunoprecipitated DNAs from the test and control samples were submitted to Illumina TruSeq library preparation, and ChIP-enriched DNA was sequenced using the Illumina HiSeq 2000 sequencing system.

### 4.8. RNA-Seq and Transcriptomic Analysis

We grew *B. melitensis* 16M biofilms in 24-well plates as described above. After 20 d of incubation, the media containing unattached planktonic bacteria were removed, and the cells were washed twice with PBS. The attached cells were scraped off the plates using a cell scraper. The samples were then transferred to a sterilized agate mortar, and liquid nitrogen was added to quickly grind the sample into a thick lysate solution. An RNA extraction kit (Takara, Beijing, China) was used for RNA isolation. For RNA-Seq, RNA purification was conducted with the MiniBEST Universal RNA Extraction kit (Takara, Beijing, China). After removal of rRNA using the RiboCop kit (Lexogen, Vienna, Austria), mRNA was used to generate the cDNA library following the MaxUp II Dual-mode mRNA Library Prep kit protocol (Hieff, China). This library was then sequenced using the HiSeq 2000 system (Illumina). We downloaded the reference genome sequence from the National Center for Biotechnology Information (NCBI) database. The raw sequencing reads were cleaned by removing low-quality reads, reads containing poly-N sequences, and adaptor sequences. HISAT40 was used to align the reads to the reference genome (GCF_000007125.1). The expression values were measured in reads per kilobase per million mapped reads (RPKM). Padj ≤ 0.05 and absolute value of log2 ratio ≥ 1 were used to identify DEGs. The GO and KEGG databases were used to analyze the pathways. All assays were performed thrice.

### 4.9. Determination of Biofilm Bacteria’s Viability by Flow Cytometry

The method for determining bacterial viability (live/dead) using flow cytometry was based on previously published methods, with some modifications [13]. Briefly, the biofilms were passed over a 50 μm filter, collected by centrifugation at 4000× *g* for 5 min, and washed twice with PBS buffer solution. The fluorescent probe Syto9/Pi was used to label bacteria in the biofilm. To distinguish bacteria in different states, we conducted a single staining test with Syto9/Pi, and selected the appropriate peak. The samples were evaluated with a flow cytometer equipped with a blue laser and bandpass filter to measure green fluorescence. Analyses were performed using biological triplicates, and the data were analyzed using FlowJo X software (TreeStar, Ashland, OR, USA).

### 4.10. Quantitative Real-Time PCR

The biofilms formed by culturing the *Brucella* mutant strain and WT strain for 20 d were gently washed three times with PBS, and RNA molecules were directly extracted. The processed samples were poured into an agate mortar that had been sterilized in advance, and liquid nitrogen was added to quickly grind the samples into a thick liquid. An RNA extraction kit was used for RNA isolation (Takara, Beijing, China). RNA molecules in the extracted samples were reverse-transcribed into cDNA. The primers are listed in Appendix A, and the qRT-PCR conditions consisted of 5 min at 95 °C for pre-incubation, followed by 35 cycles of amplification (95 °C for 40 s, 63 °C for 35 s, and 72 °C for 40 s). qRT-PCR was carried out using a QuantStudio™ 7 Flex system (Thermo Fisher Scientific, Sunnyvale, CA, USA). The samples were evaluated in triplicate and amplified in a 20 µL reaction containing 2× SYBR Premix Ex Taq II (Takara, USA). All assays were performed thrice.

### 4.11. Protein Purification and EMSAs

As part of the FtcR protein recombination study, the resulting product of *ftcR* was cloned into the pET-30a vector by cloning the NdeI and HindIII sites, and then transformed into the *E. coli DE3* strain (Tiangen, Beijing, China). Purification of FtcR was achieved using a His-tag purification resin (Tiangen, China). In order to prepare *ftcR* and *betI* DNA probes, two-step PCR was used to label the probes with biotin. Briefly, amplification was carried out with the primer pair EMSA F/R, which included probe primer oligonucleotides, to amplify 200 bases of the target-sequence-containing motifs. Biotin-labeled probe DNA was generated by amplification of the resulting products using a probe primer labeled with biotin. All DNA probes were amplified by PCR using the primers listed in Appendix A. The probes were mixed with various amounts of proteins and incubated at room temperature for 30 min. The samples were analyzed using 6% polyacrylamide gel electrophoresis. The gels were treated with a DNA dye for 10 min and photographed with a gel imaging system (Bio-Rad, Hercules, CA, USA).

### 4.12. β-Galactosidase Activity Assay

The reporter pftcR-lacZ fusion pBBCmpftcR-lacZ was carried out as described previously [22]. To construct the reporter fusion pBBCmp*ftcR*-*lacZ*, an amplicon containing the *ftcR* promoter was amplified using the primers p2c2Xamont and p2c2Baval, containing XbaI and BamHI sites from genomic DNA of the *B. melitensis* 16M strain. The PCR products were subcloned into pGEM-T Easy vectors (Promega) and then inserted into pBBCm-*lacZ* vectors [40] in frame with the *lacZ* reporter gene, generating pBBCmp*ftcR*-*lacZ*. Assays of galactosidase were conducted according to Miller’s protocol [67], as described by Fretin et al. [40].

### 4.13. Statistical Analysis

Statistical analysis was performed using GraphPad Prism software (GraphPad, Inc., La Jolla, CA, USA). Data from multiple groups were analyzed by one-way ANOVA with Dunnett’s multiple comparison test. Comparisons between two groups were performed with an unpaired Student’s *t*-test. The specific tests used for analysis are indicated in the corresponding figure legends. For bacterial replication, a non-parametric Kruskal–Wallis test with Dunn’s multi-comparison statistical analysis was performed, as the values did not follow a normal distribution. Statistical significance was defined as *p* ≤ 0.05.

## Figures and Tables

**Figure 1 ijms-23-09905-f001:**
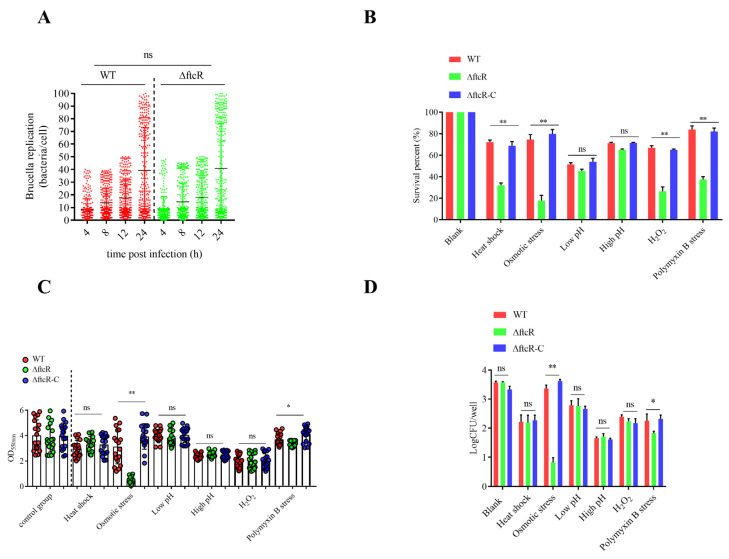
Intracellular survival and stress resistance of Δ*ftcR* during the life cycle of *Brucella melitensis* 16M: (**A**) Intracellular survival of GFP-expressing WT and Δ*ftcR* strains in RAW264.7 cells at 4, 8, 12, and 24 h. Values are means ± SD from at least three independent experiments, and were assessed using a nonparametric Kruskal–Wallis test with Dunn’s multi-comparison statistical analysis. (**B**) The survival of WT, Δ*ftcR*, and Δ*ftcR*-C strains under stress conditions. The WT, Δ*ftcR*, and Δ*ftcR*-C strains were grown in Brucella Broth to the stationary phase, after which the cells were washed and incubated under different stress growth conditions. Viable cell counts were recorded after 0.5 h of challenge. (**C**) The biofilms of WT, Δ*ftcR*, and Δ*ftcR*-C were treated with different stress conditions for 20 d, and their biomass was quantified using a crystal violet (CV) assay. Each point represents an independent well. (**D**) Survival of WT and Δ*ftcR* cells in biofilms grown under different stress conditions. Error bars represent standard error (*n* ≥ 3). * *p* ≤ 0.05, ** *p* ≤ 0.01, unpaired Student’s *t*-test. ns, not significant.

**Figure 2 ijms-23-09905-f002:**
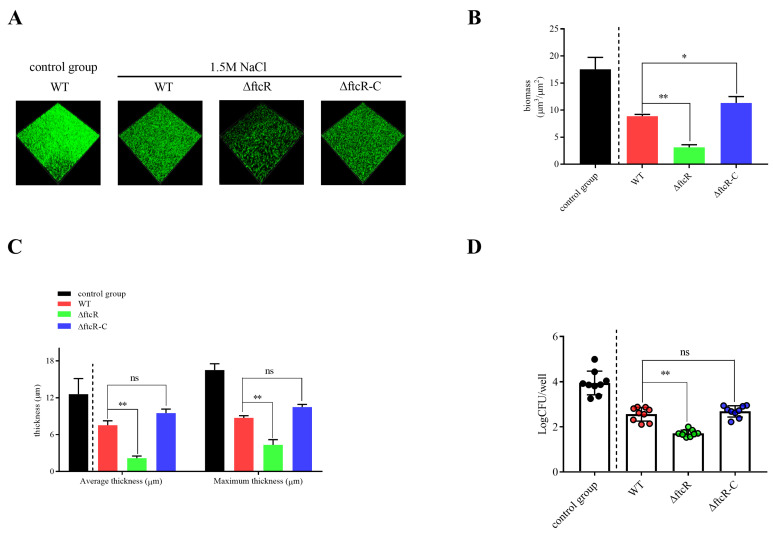
Biofilm resistance in WT, Δ*ftcR*, and Δ*ftcR*-C strains in response to hyperosmotic stress: (**A**) Live confocal imaging of stacks of WT, Δ*ftcR*, and Δ*ftcR*-C biofilms grown for 20 d in Brucella Broth medium with or without 1.5 M NaCl. Scale bars: 20 µm. (**B**,**C**) Confocal images were subjected to quantitative analysis using the Comstat2 program to determine the biomass of the biofilms of WT, Δ*ftcR*, and Δ*ftcR*-C (**B**) and their average and maximum thickness (**C**). (**D**) Survival of WT, Δ*ftcR*, and Δ*ftcR*-C in biofilms grown with or without 1.5 M NaCl treatment. Error bars represent standard error (*n* ≥ 3). * *p* ≤ 0.05, ** *p* ≤ 0.01, unpaired Student’s *t*-test. ns, not significant.

**Figure 3 ijms-23-09905-f003:**
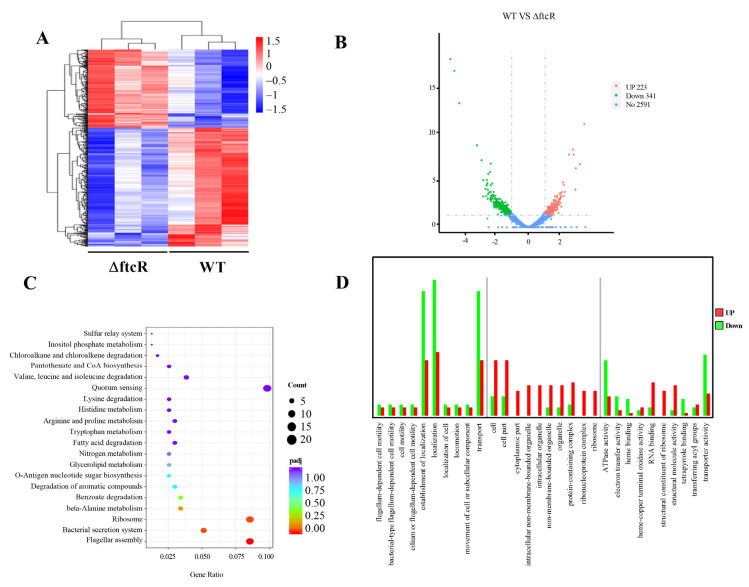
Expression of differentially expressed genes (DEGs) between the biofilms of the WT and ∆*ftcR* strains: (**A**) Heatmap showing the expression levels of DEGs between the biofilms of WT and ∆*ftcR*. (**B**) Volcano plot of expressed genes between the biofilms of WT and ∆*ftcR*. The red, green, and blue colors denote upregulated, downregulated, and non-regulated genes, respectively, based on the following criteria: absolute log2 (fold change) ≥ 1, and adjusted *p* (Padj) ≤ 0.01. (**C**) Kyoto Encyclopedia of Genes and Genomes (KEGG) analysis of the number and function of differentially expressed genes. The rich factor represents the ratio of DEG numbers annotated with this pathway term to all gene numbers annotated with this pathway term. A greater rich factor indicates a greater degree of pathway enrichment. Padj represents the corrected *p*-value, and ranges from 0 to 1; a lower value indicates greater pathway enrichment. (**D**) Analysis of Gene Ontology (GO) terms.

**Figure 4 ijms-23-09905-f004:**
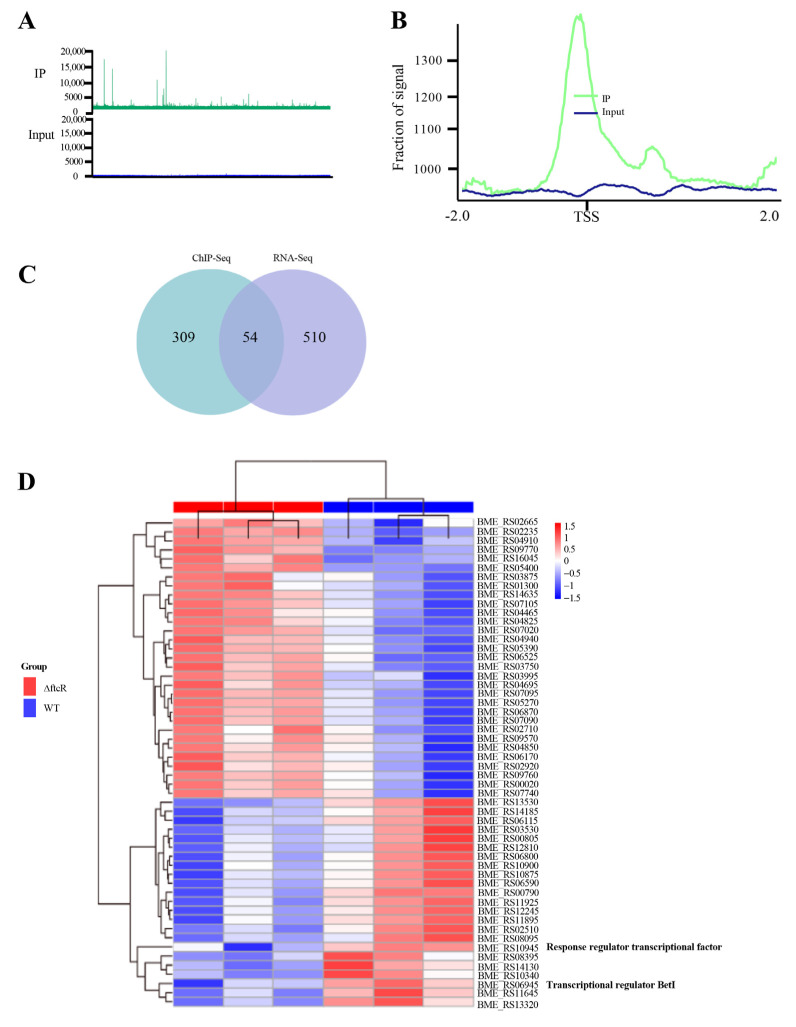
Genome-wide identification of the FtcR regulons: (**A**) A representative image of the FtcR ChIP-Seq read alignments to the *B. melitensis* 16M genome, visualized as ChIP-Seq peaks using the Integrated Genome Viewer software. Peak height correlates with the number of reads from the input (blue) or the IP (green) at the corresponding positions of the *B. melitensis* 16M genome. (**B**) Density plot of FtcR ChIP-Seq reads at 4 kb genomic regions centered at peak summits. (**C**) Venn diagram of intersecting genes observed to be differentially expressed in RNA-Seq and genes identified as directly bound by FtcR in ChIP-Seq analyses. A total of 54 genes present at the intersection of these three datasets were defined as the regulons of FtcR. (**D**) A heatmap illustrating the 54 differentially expressed regulons of FtcR between the biofilms of WT and ∆*ftcR*.

**Figure 5 ijms-23-09905-f005:**
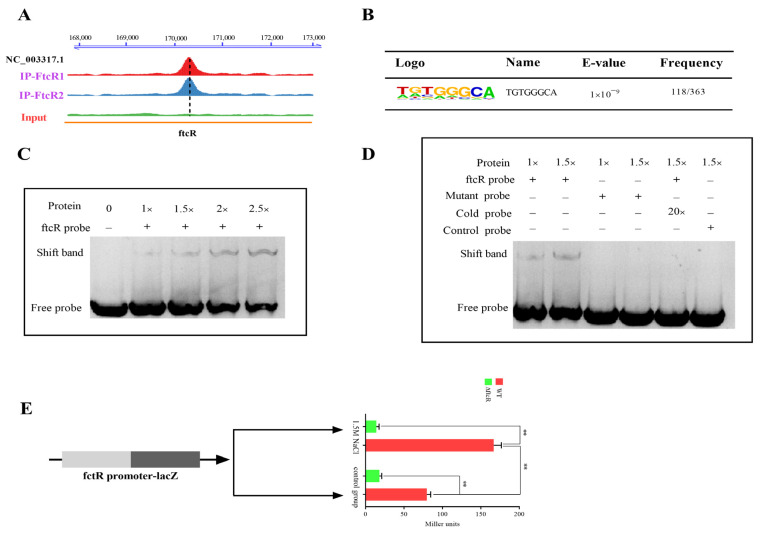
FtcR binds to the promoter region of *ftcR* in vivo and in vitro: (**A**) FtcR-enriched sequence reads located within the promoter regions of *ftcR*, with annotated genes shown at the bottom—ChIP1 and ChIP2—from two independent experimental replicates. Scale bar: 1 kb. (**B**) Conserved motifs identified by FtcR ChIP-Seq. **(****C)** EMSA for FtcR binding to the TGTGGGCA motif in the *ftcR* promoter fragments, labeled by biotin. (**D**) Specificity and competition assays. The TGTGGGCA motif was mutated into TGTCCTTG to test for sequence specificity. The unlabeled cold *ftcR* promoter region was tested for competition with the biotin-labeled *ftcR* promoter. The unlabeled cold *ftcR* promoter region inhibited the binding of FtcR to the labeled promoter. The symbols “−“ and “+” indicate the absence and presence, respectively, of the corresponding proteins or probes. Unrelated probes were used as negative controls. Results are representative of at least three independent experiments. (**E**) The effect of FtcR on the expression of the cognate promoter was detected by β-galactosidase assays using an *ftcR* promoter*–lacZ* transcriptional fusion reporter plasmid. Data represent the mean and standard deviation of *n* = 3 (independent biological replicates). The asterisks indicate significant differences (** *p* < 0.01) based on one-way ANOVA followed by Tukey’s post hoc test of honestly significant differences (two-tailed).

**Figure 6 ijms-23-09905-f006:**
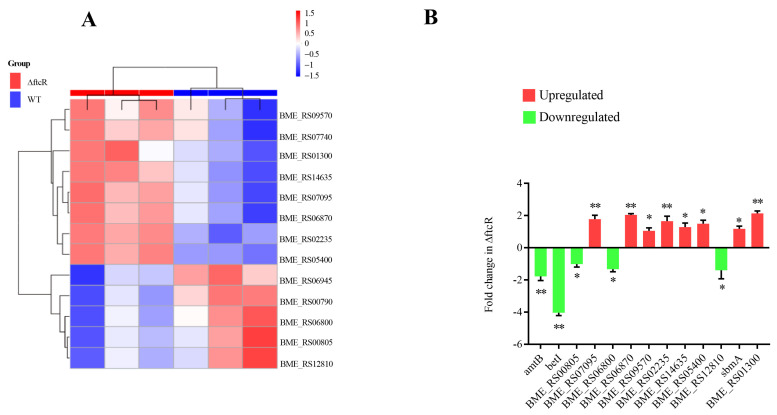
Differentially expressed genes (DEGs) were evaluated by quantitative reverse-transcription PCR (qRT-PCR) assays between the biofilms of the WT and ∆*ftcR* strains: (**A**) Heatmap showing the expression levels of 13 transporter and binding proteins between the biofilms of the WT and ∆*ftcR* strains. (**B**) The 13 transporter and binding proteins’ expression levels were further detected by qRT-PCR. Error bars represent standard error (*n* ≥ 3). * *p* ≤ 0.05, ** *p* ≤ 0.01, unpaired Student’s *t*-test.

**Figure 7 ijms-23-09905-f007:**
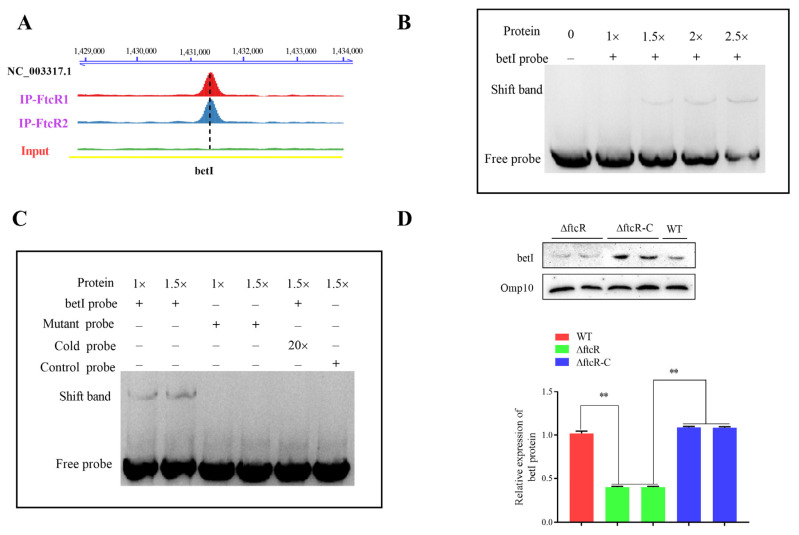
FtcR binds to the promoters of *betI* in vivo and in vitro: (**A**) FtcR-enriched sequence reads located within the promoter regions of *betI*, with annotated genes shown at the bottom—ChIP1 and ChIP2—from two independent experimental replicates. Scale bar: 1 kb. (**B**) EMSA for FtcR binding to the TGTGGGCA motif in the *betI* promoters’ fragments, labeled by biotin. (**C**) Specificity and competition assays. The TGTGGGCA motif was mutated into TGTCCTTG to test for sequence specificity. The unlabeled cold *betI* promoter region was tested for competition with biotin-labeled *betI* promoter. The unlabeled cold *betI* promoter region inhibited the binding of FtcR to the labeled promoter. The symbols “−” and “+” indicate the absence and presence, respectively, of the corresponding proteins or probes. Unrelated probes were used as negative controls. Results are representative of at least three independent experiments. (**D**) Expression of *betI* was detected by WB. Protein expression was evaluated by measuring the mean gray values of the Western blots using ImageJ. Data represent the mean and standard deviation of *n* = 3 (independent biological replicates). ** *p* ≤ 0.01, unpaired Student’s *t*-test.

**Figure 8 ijms-23-09905-f008:**
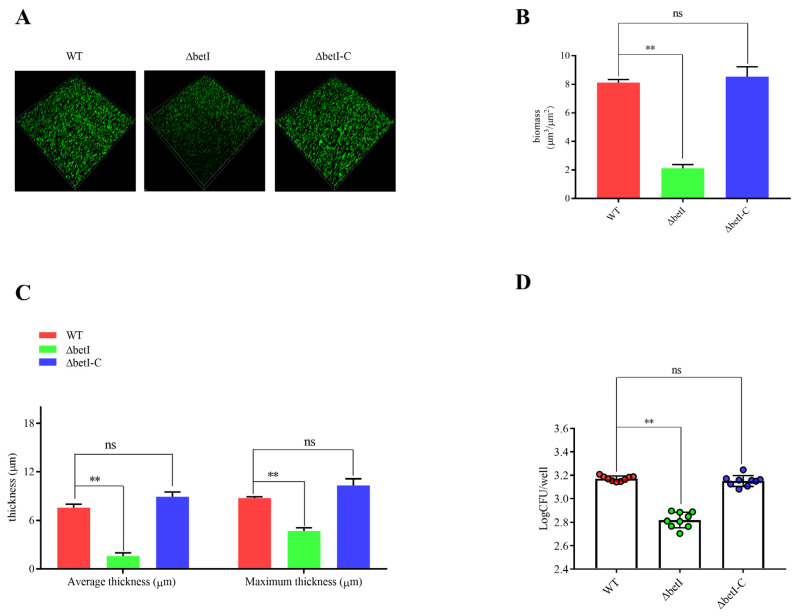
Biofilm resistance of WT, Δ*betI*, and Δ*betI*-C in response to hyperosmotic stress: (**A**) Live confocal imaging of stacks of WT, Δ*betI*, and Δ*betI*-C biofilms grown for 20 d in Brucella Broth medium with 1.5 M NaCl. Scale bars: 20 µm. (**B**,**C**) Confocal images were subjected to quantitative analysis using the Comstat2 program to determine the biofilm biomass (**B**) and the average and maximum biofilm thickness (**C**). (**D**) Survival of bacteria in biofilms grown under 1.5 M NaCl treatment. Error bars represent standard error (*n* ≥ 3). ** *p* ≤ 0.01, unpaired Student’s *t*-test. ns, not significant.

**Figure 9 ijms-23-09905-f009:**
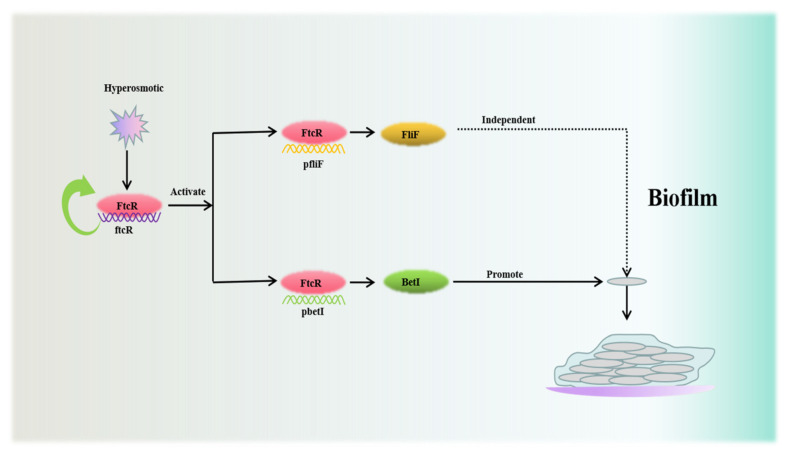
Regulatory mechanism of the transcriptional factor FtcR on biofilm formation during hyperosmotic stress: Under hyperosmotic stress, the transcriptional factor FtcR plays an essential role in autoregulating its expression and the osmotic-stress-response regulator by directly binding to the conserved TGTGGGCA motif upstream of *betI*, which is involved in controlling *B. melitensis* 16M biofilm formation when challenged by osmotic stress. However this process is independent of its flagellar target gene, *fliF*.

## Data Availability

The ChIP-Seq and RNA-Seq data generated in this study were submitted to SRA under the GenBank accession numbers PRJNA871752 and PRJNA871391, respectively.

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
