# Peer review of "The Flagellar Transcriptional Regulator FtcR Controls Brucella melitensis 16M Biofilm Formation via a betI-Mediated Pathway in Response to Hyperosmotic Stress"

_ijms, 2022, doi:10.3390/ijms23179905_

Round 1
Reviewer 1 Report
The present work investigated the role of flagellar transcription factor (FtcR) to control the biofilm formation of B. melitensis 16M in response to hyperosmotic stress. The authors also identify how the regulatory cascade of the hyperosmotic stress response occurs through the regulation of the stress response regulator BetI. The article is well written and easy to follow.
I have not observed any major points, however, some minor aspects should be revised to improve the article :
- In line 62, the cited article 18 does not refer to Brucella but to Staphylococcus. Please change the reference or rephrase the sentence in a more correct manner.
- Figure 2A shows the WT image twice with 1.5 M NaCl. I assume that the first image refers to the control condition without NaCl.
Author Response
Thank you for reviewing our manuscript and for offering such constructive feedback. Furthermore, we appreciate your recommending our paper to the editor. The updates we made in relation to your comments have greatly improved the quality of our paper. The responses to each of your comments are as follows:
First of all, English language and style have edited by a native speaker.
1.- In line 62, the cited article 18 does not refer to Brucella but to Staphylococcus. Please change the reference or rephrase the sentence in a more correct manner.
Our response: Thank you for pointing it out. Corrected as suggested. We have corrected the cited article errors caused by manuscript typesetting, and adjusted the cited article 18 as “Flury, D.; Behrend, H.; Sendi, P.; von Kietzell, M.; Strahm, C. Brucella melitensis prosthetic joint infection. J. Bone Jt. infect. 2017, 2, 136–142.”.
2.- Figure 2A shows the WT image twice with 1.5 M NaCl. I assume that the first image refers to the control condition without NaCl.
Our response: Thank you for pointing it out. Corrected as suggested. We have removed the line for 1.5M NaCl on the left most (WT) figure.
Reviewer 2 Report
Summary: Accept with minor corrections.
This manuscript seeks to address the role of ftcr in bacterial response to hyperosmotic stress. The study is well designed and executed, and this manuscript is well suited for IJMS. There are a few minor questions/comments to be addressed this manuscript:
1. Please keep a consistent color scheme throughout the manuscript for each strain (such as that used in the majority of this manuscript: red for WT, green for ΔftcR, blue for ΔftcR-C). Figs. 1b, 5e, and 6d should be changed accordingly.
2. Fig 2a should not have the line for 1.5M NaCl on the left most (WT) figure
3. In addition to autoregulation, does any of the other ftcR targets identified in Fig. 4d bind to the promoter region of ftcR to regulate its transcription?
4. Section 2.5 seems out of place in this manuscript. Is there a better way to incorporate it?
5. Section 2.6: “We found that the majority of the genes differentially regulated by FtcR in biofilm cells 279 encoded transport and binding proteins, suggesting FtcR mediates the regulation of 280 transport during the development of B. melitensis 16M biofilms”. Is this true only for biofilm bacteria and not for bacteria in other stages?
6. Regulation of betI gene expression in ΔftcR (supplementary figure S5B) should be a main figure in Fig. 6 as it is an important result.
7. Is the expression of betI restored in ΔftcR-C strain?
Author Response
Thank you for reviewing our manuscript and for offering such constructive feedback. Furthermore, we appreciate your recommending our paper to the editor. The updates we made in relation to your comments have greatly improved the quality of our paper. The responses to each of your comments are as follows:
- Please keep a consistent color scheme throughout the manuscript for each strain (such as that used in the majority of this manuscript: red for WT, green for ΔftcR, blue for ΔftcR-C). Figs. 1b, 5e, and 6dshould be changed accordingly.
Our response: Thank you for pointing it out. Corrected as suggested. We have adjusted the color scheme to keep a consistent color scheme throughout the manuscript for each strain.
- Fig 2a should not have the line for 1.5M NaCl on the left most (WT) figure
Our response: Thank you for pointing it out. Corrected as suggested. We have deleted the line for 1.5M NaCl on the left most (WT) figure.
- In addition to autoregulation, does any of the other ftcR targets identified in Fig. 4d bind to the promoter region of ftcR to regulate its transcription?
Our response: Thank you for pointing it out. This is a question worth considering and researching. In our study, we mainly elucidate the target genes of the flagellar transcriptional regulator FtcR. In addition to their own self-regulation, the proteins encoded by the target genes we identified may also regulate ftcR, but further verifications may be required, such as screening sense the proteins encoded by the target gene of interest, using Chip-Seq, EMSA and RNA-Seq again. This is exactly what we are going to do in the future.
- Section 2.5 seems out of place in this manuscript. Is there a better way to incorporate it?
Our response: Thank you for pointing it out. Corrected as suggested. We have incorporated Section 2.5 with Section 2.4 as “2.4. Transcriptional Autoregulatory Properties of FtcR And Independent of FliF”.
5.Section 2.6: “We found that the majority of the genes differentially regulated by FtcR in biofilm cells encoded transport and binding proteins, suggesting FtcR mediates the regulation of transport during the development of B. melitensis 16M biofilms”. Is this true only for biofilm bacteria and not for bacteria in other stages?
Our response: Thank you for your question. Biofilm development is a complex process, compared to planktonic bacteria, biofilms have unique transcriptional and expressive programs [1, 2]. We screened these genes differentially regulated by FtcR in biofilm cells using transcriptomics, and studied the key functions of some important genes in the biofilm stage, taking into account the differences in regulation and function of genes at different stages [3], we are working on the question you mentioned about the functional differences of genes at different stages (e.g. biofilm and planktonic states).
The references are as follows:
1.Tang, T.; Xu, Y.; Wang, J.; Tan, X.; Zhao, X.; Zhou, P.; Kong, F.; Zhu, C.; Lu, C.; Lin, H. Evaluation of the differences between biofilm and planktonic Brucella abortus via metabolomics and proteomics. Funct. Integr. Genomics 2021, 21, 421-433.
2.Tang, T.; Chen, G.; Guo, A.; Xu, Y.; Zhao, L.; Wang, M.; Lu, C.; Jiang, Y.; Zhang, C. Comparative proteomic and genomic analyses of Brucella abortus biofilm and planktonic cells. Mol. Med. Rep. 2020, 21, 731–743.
3.Kleinman, C.L.;Sycz, G.;Bonomi, H.R.; Rodríguez, R.M.; Zorreguieta, A.; Sieira, R. ChIP-seq analysis of the LuxR-type regulator VjbR reveals novel insights into the Brucella virulence gene expression network. Nucleic Acids Res. 2017, 45, 5757–5769.
- Regulation of betI gene expression in ΔftcR (supplementary figure S5B) should be a main figure in Fig. 6 as it is an important result.
Our response: Thank you for your suggestion. Corrected as suggested. In order to maintain high resolution, we moved the “supplementary figure S5B” to the main figure as new “Figure 5”. (We did not merge supplementary figure S5B into Figure 6 because doing so would result in a low resolution).
- Is the expression of betI restored in ΔftcR-C strain?
Our response: Thank you for your question. The the expression of betI was restored in ΔftcR-C strain. For completeness and logic of the manuscript, we redo this part of the experiment, adding the expression of betI in the ΔftcR-C strain. The result showed that the expression of betI was restored in ΔftcR-C strain in the Figure 7D.
Reviewer 3 Report
In this study, the authors demonstrate the involvement of the flagellar transcriptional regulator protein, FtcR, in B. melitensis 16M biofilm tolerance of hyperosmotic stress. This work might help us to understand the biofilm-specific tolerance and facilitate development of counteracting strategies. The authors did a comprehensive study to investigate the FtcR which mediates biofilm formation under hyperosmotic stress. The experiment design is appropriate and lots of work are well-done. If possible, the English language and style are required to be edited by a native speaker. Some comments:
1. Although ΔftcR-C is known as the ftcR complemented strain in the following section of the manuscript, the authors should explain ΔftcR-C when it initially appeared in the text (line 110). And it’s better to briefly explain how to construct this complemented strain.
2. Figure 2A, the most left panel should be wild-type without 1.5M NaCl treatment.
3. What is “REC” in line 174.
4. “ftcR-lacZ plasmid” or figure 5E, the cassette should be “fctR promoter-lacZ”? The current one looks like the fusion of FtcR protein and LacZ protein.
5. Figure 4A resolution or size is not good. It’s difficult to see any details.
6. It’s not clear about “ftcr probe, mutant probes, specific cold probe competitors (unlabeled probe), and a control probe” in figure 5. More explanations are needed in the text or legend.
7. Replace “using qRT-PCR” by “as determined by qRT-PCR” (line 288).
8. It would be very helpful for the readers to get a clear take-home-message if there is a schematic model about ftcR regulation signal at the end of the manuscript.
Author Response
Thank you for reviewing our manuscript and for offering such constructive feedback. Furthermore, we appreciate your recommending our paper to the editor. The updates we made in relation to your comments have greatly improved the quality of our paper. The responses to each of your comments are as follows:
- If possible, the English language and style are required to be edited by a native speaker.
Our response: Thank you for pointing it out. Corrected as suggested. English language and style have edited by a native speaker. The English editing certificate of this manuscript has been uploaded as an attachment.
- Although ΔftcR-C is known as the ftcR complemented strain in the following section of the manuscript, the authors should explain ΔftcR-C when it initially appeared in the text (line 110). And it’s better to briefly explain how to construct this complemented strain.
Our response: Thank you for pointing it out. Corrected as suggested. we answer this comments in two parts of A and B:
Part A: Although ΔftcR-C is known as the ftcR complemented strain in the following section of the manuscript, the authors should explain ΔftcR-C when it initially appeared in the text (line 110).
Our response: Thank you for pointing it out. Corrected as suggested. We have revised this parts according to the comments, that is, “Results show that the survival rate of ∆ftcR decreased under heat shock, osmotic stress, H2O2, and polymyxin stress conditions compared to that of the WT strain (Figure 1B), but was restored partly or fully in ∆ftcR complemented strain, ∆ftcR-C”.
Part B: And it’s better to briefly explain how to construct this complemented strain.
Our response: Corrected as suggested. We have added a description of the construction of the complement strain in “Materials and Methods” according to the comments, that is, “The ftcR, fliF, and betI complemented strains (∆ftcR-C, ∆fliF-C, and ∆betI-C, respectively) were acquired as described previously, the complementation vector pBBR1MCS4, a plasmid that could replicate in Brucella, was introduced into electro-competent cells of ∆ftcR, ∆fliF and ∆betI via electroporation”.
- Figure 2A, the most left panel should be wild-type without 1.5M NaCl treatment.
Our response: Thank you for pointing it out. Corrected as suggested. We have deleted the line for 1.5M NaCl on the left most (WT) figure.
- What is “REC” in line 174.
Our response: Thank you for pointing it out. “REC” is the receiver (REC) domain. We have fully described in the manuscript, that is “FtcR is predicted to function as a DNA-binding response regulator that contains receiver (REC) and winged-helix (wHTH) domain”.
- “ftcR-lacZ plasmid” or figure 5E, the cassette should be “fctR promoter-lacZ”? The current one looks like the fusion of FtcR protein and LacZ protein.
Our response: Thank you for pointing it out. Corrected as suggested. We have changed the “ ftcR-lacZ plasmid” or “figure 5E” as “fctR promoter-lacZ”.
- Figure 4A resolution or size is not good. It’s difficult to see any details.
Our response: We apologize for the poor resolution. We have adjusted the presentation of Figure 4A to better capture valuable details.
- It’s not clear about “ftcr probe, mutant probes, specific cold probe competitors (unlabeled probe), and a control probe” in figure 5. More explanations are needed in the text or legend.
Our response: Thank you for pointing it out. Corrected as suggested. “ftcr probe” refers to: Electrophoretic mobility shift assay (EMSA) for FtcR binding to the TGTGGGCA motif in the ftcR promoters fragments, which was labeled by biotin. Mutated probe refers to: The TGTGGGCA motif was mutated into TGTCCTTG to test for sequence specificity. Specific cold probe competitors refers to: Unlabeled cold ftcR promoter region was tested for competition with biotin-labeled ftcR promoter. Unlabeled cold ftcR promoter region inhibits the binding of ftcR to the labeled promoter; and unrelated probe were used as the negative controls. All the above explanations has been supplemented in the legend of Figure 5 and 7.
- Replace “using qRT-PCR” by “as determined by qRT-PCR” (line 288).
Our response: Thank you for your suggestion. Corrected as suggested. We have revised this sentence as “The expressions of these genes, as determined by qRT-PCR, coincided with the RNA-Seq results (Figure 5B)”.
- It would be very helpful for the readers to get a clear take-home-message if there is a schematic model about ftcR regulation signal at the end of the manuscript.
Our response: Thank you for your suggestion. Corrected as suggested. We have added the “Figure 8. Regulatory mechanism of the FtcR transcriptional factor on biofilm formation during hyperosmotic stress” at the end of the manuscript.

Round 2
Reviewer 3 Report
All comments are addressed.
Can be published when a minor typo is modified: in figure 9, it should be "activate" instead of "activaye".